# Modeling and Forecasting Dead-on-Arrival in Broilers Using Time Series Methods: A Case Study from Thailand

**DOI:** 10.3390/ani15081179

**Published:** 2025-04-20

**Authors:** Chalita Jainonthee, Panneepa Sivapirunthep, Pranee Pirompud, Veerasak Punyapornwithaya, Supitchaya Srisawang, Chanporn Chaosap

**Affiliations:** 1Faculty of Veterinary Medicine, Chiang Mai University, Chiang Mai 50100, Thailand; chalita.j@cmu.ac.th (C.J.); veerasak.p@cmu.ac.th (V.P.); 2Veterinary Public Health and Food Safety Centre for Asia Pacific (VPHCAP), Faculty of Veterinary Medicine, Chiang Mai University, Chiang Mai 50100, Thailand; 3Research Center for Veterinary Biosciences and Veterinary Public Health, Faculty of Veterinary Medicine, Chiang Mai University, Chiang Mai 50100, Thailand; s.supitchaya45@gmail.com; 4Department of Agricultural Education, Faculty of Industrial Education and Technology, King Mongkut’s Institute of Technology Ladkrabang, Bangkok 10520, Thailand; panneepa.si@kmitl.ac.th; 5Sun Group Company, Jathujak, Bangkok 10900, Thailand; pirompud@gmail.com

**Keywords:** antibiotic-free broiler production, dead-on-arrival, time series forecasting, poultry welfare, SARIMA, NNAR, TBATS, ETS, XGBoost, seasonal pattern

## Abstract

Antibiotic-free broiler production is growing in popularity due to its health and welfare benefits. However, broilers raised without antibiotics are more vulnerable to stress and disease, particularly during transportation to slaughterhouses. A common issue in the poultry industry is “dead on arrival,” or birds that do not survive the trip. High percentages of dead on arrival not only reduce profit but also raise concerns about animal welfare. In this study, we used data collected from a large poultry producer in Thailand over seven years to examine patterns in broiler mortality during transport. We applied forecasting models to predict when higher mortality might occur, helping producers plan ahead and prevent losses. Our results showed that some forecasting methods were more accurate than others in identifying seasonal trends in mortality. This type of predictive tool can help producers improve animal care, reduce unnecessary losses, and support sustainable farming practices.

## 1. Introduction

Since 2018, a prominent broiler producer in Thailand and exporter to the European market has adopted an antibiotic-free (ABF) production system to ensure high-quality poultry products while supporting sustainable and ethical farming practices [1]. The implementation of ABF production, although beneficial from a consumer and regulatory standpoint, introduces challenges such as increased disease susceptibility and higher mortality rates during production and transport [2]. Meeting European welfare standards requires stricter management and monitoring practices, particularly when rearing broilers without antibiotic intervention [3].

One of the critical indicators in broiler production is the percentage of dead-on-arrival birds (%DOA), which reflects preslaughter mortality [4,5,6]. This parameter is essential not only for economic and operational reasons but also as a benchmark of animal welfare [1,7,8]. %DOA accounts for broilers that die during the catching, loading, and transportation process before slaughter, often due to stress, poor ventilation, temperature extremes, or rough handling [9,10]. A high %DOA increases economic losses through reduced revenue, disrupted logistics, and decreased processing efficiency [7,11], and it also signals potential welfare violations that may jeopardize compliance with international animal welfare standards.

Analyzing time series data of %DOA allows for the identification of seasonal and temporal patterns that support proactive intervention strategies. For example, recognizing periods of elevated DOA risk allows producers to adjust loading densities, transport schedules, or handling procedures accordingly. However, conventional management practices often rely on manual evaluation and retrospective summaries, which may be limited in scope, accuracy, and scalability. In contrast, time series forecasting techniques provide a systematic, data-driven approach to detect trends and anticipate future outcomes with greater precision [12].

Several studies have demonstrated the successful application of time series models across agriculture, veterinary, and public health domains for predicting outcomes such as agricultural production volumes [13,14,15], disease patterns in animals and humans [16,17,18,19], and microbial risks [20,21]. These models are particularly effective at capturing temporal dependencies, identifying seasonal patterns, and modeling non-linear trends in complex biological and production systems. Their application has supported data-driven planning and risk mitigation strategies in areas such as animal health, food production, and public health.

Although time series models are increasingly applied in animal production research, few studies have specifically focused on forecasting %DOA in poultry, particularly within ABF broiler production systems. To address this gap, the present study evaluated the use of widely applied time series forecasting models for predicting %DOA in broilers raised under the ABF program. The selected models included Seasonal AutoRegressive Integrated Moving Average (SARIMA), Neural Network AutoRegressive (NNAR), Trigonometric Box-Cox ARMA Trend Seasonal (TBATS), Exponential Smoothing State Space (ETS), and Extreme Gradient Boosting (XGBoost). Given that previous studies have demonstrated a seasonal pattern in %DOA data in Thailand [1], the SARIMA, TBATS, and ETS models were chosen for evaluation due to their ability to account for seasonality in the forecasting process [22,23,24,25,26]. Additionally, NNAR was included as a candidate model to capture potential nonlinear patterns in the data [16,27,28]. XGBoost was also examined due to its capacity to manage complex and nonlinear data structures, making it a suitable alternative for enhancing forecasting performance [29].

The objectives of this study were to (i) analyze %DOA data from 2018 to 2024 to identify underlying trends and seasonal patterns, (ii) develop and evaluate forecasting models, and (iii) predict %DOA values for 2025. The findings offer practical insights to enhance animal welfare, reduce production losses, and support long-term operational planning. This approach also provides a framework for integrating forecasting models into precision poultry farming and supply chain optimization.

## 2. Materials and Methods

### 2.1. Data Collection

This study utilized daily records of the percentage of dead-on-arrival (%DOA) broilers from a large commercial broiler producer in Thailand that has operated under an ABF program since 2018. The primary outcome variable was the %DOA per transport truckload (n = 127,578), calculated by dividing the number of DOA broilers by the total number of broilers transported in each truckload. Daily %DOA records, collected from January 2018 to December 2024, were aggregated into monthly averages to facilitate time series analysis, resulting in a dataset spanning seven years. Prior to aggregation, the daily %DOA data were screened for extreme outliers that could potentially distort the analysis. Outliers were defined as values that substantially deviated from the expected range and were likely attributable to exceptional events [30], such as mechanical failures during transportation. These data points were excluded from further analysis to maintain data integrity and analytical consistency.

### 2.2. Rearing, Handling, and Transportation Protocols

Broilers were raised under the ABF management system in evaporatively cooled housing. Antibiotics were not administered at any point in the production cycle, including through feed, water, or injection. Instead, birds were supplemented with chemical coccidiostats, probiotics, phytogenic additives, and essential oils.

Prior to transportation, feed and water were withdrawn. Broilers were manually captured and placed into transport crates, with 4 to 10 birds per crate, ensuring that each crate did not exceed a total weight of 18 kg. Each transport truck carried 495 crates. Upon arrival at the slaughterhouse, trucks were moved to a holding area equipped with fans and evaporative cooling systems to maintain bird welfare. After a holding time of 30 to 120 min, the trucks were moved to the unloading zone, and the crates were manually removed. Broilers were then transferred by hand from the crates to the processing line. The number of DOA broilers was recorded during ante-mortem inspection by quality control personnel, and these values were used to calculate %DOA per truckload.

### 2.3. Determination of Trends, Seasonal Patterns, and Residual Components in Actual %DOA

In time series analysis, it is essential to assess whether the data are stationary before conducting further modeling. Time series data typically consist of four main components: trend, seasonality, cyclic variations, and random (residual) fluctuations [12]. In this study, time series decomposition was applied to isolate and visualize the underlying trend, seasonal patterns, and residual components in the %DOA dataset throughout the entire study period.

The decomposition was performed on monthly average %DOA data from January 2018 to December 2024. Both seasonal and non-seasonal characteristics were explored, and analytical approaches were adjusted based on the presence or absence of seasonality.

To formally assess the presence of a seasonal component in the %DOA data, the Friedman rank sum test was performed [31,32]. The analysis was conducted using R software (version 4.4.3) [33]. In addition to the statistical test, a circular plot was created to visually illustrate seasonal variations throughout the year, using the “ggplot2” package.

### 2.4. Time Series Models for Determining and Forecasting %DOA

To forecast future %DOA in broilers raised under the ABF program, a set of five time series models was applied. These models were selected to represent a diverse range of forecasting techniques, including classical statistical approaches, neural networks, and tree-based machine learning algorithms. The models included SARIMA, NNAR, TBATS, ETS, and XGBoost.

By incorporating both statistical and machine learning-based methods, this study aimed to comprehensively evaluate each model’s capacity to capture key temporal features in the %DOA data, including trends, seasonality, and nonlinear patterns. The forecasting performance of each model was compared to determine the most effective and practical approach for application in commercial broiler production under ABF conditions.

#### 2.4.1. Seasonal AutoRegressive Integrated Moving Average (SARIMA) Model

The SARIMA model is a widely used extension of the traditional ARIMA framework for analyzing time series data with recurring seasonal patterns. It incorporates non-seasonal components, including autoregressive (AR), differencing (I), and moving average (MA) terms, as well as seasonal components such as seasonal autoregressive, seasonal differencing, and seasonal moving average terms. These additional terms allow SARIMA to effectively model seasonal dependencies and fluctuations within the data [18,34].

The model is typically expressed as *SARIMA(p,d,q)(P,D,Q)[s]*, where *p*, *d*, and *q* refer to the non-seasonal autoregressive order, differencing order, and moving average order, respectively. The parameters *P, D*, and *Q* represent their seasonal counterparts. The term *s* indicates the length of the seasonal cycle, such as *s* = 12 for monthly data with annual seasonality.

The general form of the SARIMA model is expressed as follows [22,23]:(1)$$\varphi \left(X\right)\Phi \left(X^{12}\right)\Delta^{d}\Delta_{12}^{D}Z_{t}=\theta \left(X\right)\Theta \left(X\right)a_{t}$$
where $\varphi \left(X\right)$ = non-seasonal autoregressive operator of order *p*, $\theta \left(X\right)$ = non-seasonal moving average operator of order *q*, $\Phi \left(X^{12}\right)$ = the autoregressive seasonal operator of order *P*, $\Theta \left(X\right)$ = the moving averages operator of order *Q*, $\Delta^{d}$ = the operator difference, $\Delta_{12}^{D}$ = operator seasonal difference, and $a_{t}$ = white noise.

#### 2.4.2. Neural Network AutoRegressive (NNAR) Model

The NNAR model is a time series forecasting technique that leverages artificial neural networks to capture complex, nonlinear relationships within the data [35]. The NNAR model builds upon the structure of traditional autoregressive (AR) models by incorporating neural network architectures. This enhancement allows the model to better capture complex and nonlinear temporal patterns, particularly in datasets where linear models are insufficient [36].

In the NNAR model, lagged values of the time series are used as input features to a feedforward neural network. This network typically consists of three layers: an input layer composed of lagged observations, a hidden layer, and an output layer, which are connected through acyclic pathways [37]. The network is trained to learn the underlying structure of the time series and generate forecasts for future values.

In this study, the NNAR model was applied to %DOA data using lagged monthly values as inputs to forecast future %DOA values.

The general form of the NNAR model is represented as follows [16,27,28]:(2)$$y_{t}=\omega_{0}+\sum_{j=1}^{Q}\omega_{g}g\left(\omega_{oj}+\sum_{i=1}^{p}\omega_{i,j}y_{t-1}\right)+e_{t}$$
where $y_{t}$ and $\left(y_{t-i,\ldots ,}y_{t-p}\right)$ are the output and the input, $\omega_{i,j}(i=0,1,2,\ldots ,P;\;j=1,2,\ldots Q)$ and $\omega_{j}(j=0,1,2,\ldots ,Q)$ are model parameters, which are known as connection weights; the number of input nodes is represented by P, while the number of hidden nodes is indicated by Q.

#### 2.4.3. Trigonometric Box-Cox ARMA Trend Seasonal (TBATS) Model

The TBATS model is an advanced time series forecasting method designed to capture complex seasonal patterns, including multiple and non-integer seasonality. It extends the classical Box-Cox transformation and AutoRegressive Moving Average (ARMA) models by integrating several key components that enhance forecasting accuracy. These include trigonometric representations such as Fourier terms to model seasonality, the Box-Cox transformation to address heterogeneity in variance, and ARMA errors to account for short-term dynamics. The model also incorporates damping mechanisms for trends and accommodates both standard and irregular seasonal components, allowing it to effectively model multiple overlapping seasonal patterns and nonlinear behaviors. TBATS is particularly well suited for time series with intricate seasonal structures, such as those with simultaneous daily, weekly, and annual cycles, and can handle autocorrelation in residuals [24,38].

Due to its flexibility in capturing irregular seasonal fluctuations, the TBATS model is particularly suited for time series forecasting in domains where seasonal effects do not follow a strictly periodic pattern.

The mathematical formulation of the TBATS model is expressed as follows [24,25]:(3)$$y_{t}^{\left(\omega \right)}=l_{t-1}+\phi b_{t-1}+\sum_{i=1}^{T}s_{t-m_{i}}^{\left(i\right)}+d_{t}$$
where $y_{t}^{\left(\omega \right)}$ refers to the Box-Cox transformation parameter (ω), which is applied to the observation $y_{t}$ at time t, $l_{t}$ denotes the local level, ϕ is the damped trend, b is the long-run trend, T refers to the seasonal periods, and $d_{t}$ is an ARMA(p,q) process for residuals.

#### 2.4.4. Exponential Smoothing State Space (ETS) Model

The ETS model is a time series forecasting method based on exponential smoothing, designed to accommodate various forms of trend and seasonality. It is represented as ETS (error, trend, seasonality), where each component can take different forms, including additive, multiplicative, or absent [25]. The model applies weighted smoothing to past observations, allowing it to capture underlying patterns while giving greater importance to more recent data points [26].

Within the ETS framework, the error (E) component can be either additive (A) or multiplicative (M). The trend (T) and seasonal (S) components can also be additive (A), multiplicative (M), or non-existent (N). Additionally, the trend component can be dampened additively ${(A}_{d})$ or multiplicatively $\left(M_{d}\right)$ to prevent unrealistic long-term projections. By structuring the model within a state-space representation, ETS provides a flexible and interpretable approach to forecasting time series data with diverse characteristics.

The state-space formulation of the ETS model is given as follows [26]:(4)$$y_{t}=w\left(x_{t-1}\right)+r\left(x_{t-1}\right)\varepsilon_{t}$$(5)$$x_{t}=f\left(x_{t-1}\right)+g\left(x_{t-1}\right)\varepsilon_{t}$$
where w, f, and g are coefficients, while $\varepsilon_{t}$ denotes the Gaussian white noise series. The former is known as the observation that describes the relationship between the observation $x_{t-1}$ and $y_{t}$. The latter equation represents the transitional equation describing the evolution of states over time. The ETS model was performed using the *ets( )* function from the “forecast” package.

#### 2.4.5. Extreme Gradient Boosting (XGBoost) Model

Although XGBoost is not inherently designed for time series forecasting, it can be effectively applied to time series data with appropriate modifications. This machine learning model is particularly useful for capturing complex, nonlinear patterns that traditional time series models might overlook.

XGBoost operates using an ensemble of boosted decision trees, which iteratively improve predictions by minimizing errors in a sequential manner. In the context of time series forecasting, past observations are transformed into feature representations, while future values serve as the target variable. This approach allows the model to learn dependencies within the data and generate forecasts based on trends and patterns [39].

In this study, XGBoost was employed to forecast %DOA by structuring the dataset into lagged features. The objective function is given below [29]:(6)$$Obj(\theta )=\sum_{i=1}^{n}L(y_{i},{\hat{y}}_{i})+\sum_{k=1}^{K}\Omega (f_{k})$$
where $L(y_{i},{\hat{y}}_{i})$ denotes the loss function between the actual values $y_{i}$ and predicted values ${\hat{y}}_{i}$, and $\Omega (f_{k})$ is the regularization term applied to each decision tree $f_{k}$, which penalizes the complexity of the model to avoid overfitting [40].

### 2.5. Analytical and Modeling Procedure

To evaluate the forecasting performance of each model, the dataset from 2018 to 2023 was used for training, while the 2024 dataset was reserved for testing. The modeling process consisted of two stages (Figure 1). First, a model was selected from each time series approach, and its performance was validated using the test dataset. Five time series models were assessed using the test dataset by comparing their forecasted values with actual values for the same period. In the final step, the selected models were applied to the entire dataset to predict %DOA for 2025, ensuring an up-to-date forecast.

Model performance was evaluated using four standard error metrics: mean absolute error (MAE), mean absolute percentage error (MAPE), mean absolute scaled error (MASE), and root mean square error (RMSE). Lower values of these metrics indicate better model performance [26,41,42].

MAE represents the average absolute difference between predicted (${\hat{y}}_{i}$) and actual values ($y_{i}$). It provides an intuitive measure of forecasting accuracy in the same units as the target variable, with lower values indicating better performance. The equation for calculating MAE is as follows:(7)$$MAE=\frac{1}{n}\sum_{i=1}^{n}|y_{i}-{\hat{y}}_{i}|$$

MAPE expresses forecast errors as a percentage of actual values, making it useful for assessing relative forecasting accuracy. However, it may be sensitive to very small actual values, leading to disproportionately high error percentages. The equation for calculating MAPE is shown below:(8)$$MAPE=\frac{1}{n}\sum_{i=1}^{n}\frac{\left|y_{i}-{\hat{y}}_{i}\right|}{y_{i}}\times 100$$

MASE adjusts for scale differences across datasets by normalizing errors relative to a benchmark, such as the mean absolute error of a naïve forecast. It provides a consistent metric for comparing forecasting performance across different time series. The formula used to compute MASE is given below:(9)$$MASE=\frac{1}{n}\sum_{i=1}^{n}\left(\frac{\left|y_{i}-{\hat{y}}_{i}\right|}{\frac{1}{n-1}\sum_{i=2}^{n}\left|y_{i}-y_{i-1}\right|}\right)$$

RMSE is the square root of the average squared differences between predicted and actual values. Since errors are squared before averaging, RMSE penalizes larger errors more heavily than MAE, making it particularly sensitive to outliers. The following equation is used to compute RMSE:(10)$$RMSE=\sqrt{\frac{1}{2}\sum_{i=1}^{n}(y_{i}-{\hat{y}}_{i}{)}^{2}}$$

## 3. Results

### 3.1. Exploratory Time Series Analysis of %DOA (2018–2024)

The %DOA of broilers raised under the ABF program by a large commercial broiler producer in Thailand was calculated from a total of 127,578 transport truckloads recorded between 2018 and 2024. The average monthly %DOA during this period was 0.35%, with a standard error of 0.002.

To explore temporal patterns, a time series decomposition was applied to the monthly %DOA data. As shown in Figure 2, the trend component revealed a slight decline in %DOA during 2018–2019, followed by a progressive increase through 2022, peaking in early 2023, and then declining modestly toward the end of 2024. The seasonal component exhibited a clear annual cycle, with consistent peaks occurring at the beginning of each year, particularly in February, and troughs observed in the middle to later months. The lowest %DOA was typically found in September, with a secondary low in June.

To statistically validate the presence of seasonality in the %DOA data, the Friedman rank sum test further confirmed statistically significant seasonal variation (*p* < 0.01). These findings were consistent with the visual inspection of the circular seasonal plot shown in Figure 3, which illustrates repeating annual peaks and troughs in %DOA, with observable differences between years.

### 3.2. Forecasting Model Performance on Test Data (2024)

The performance of five time series models, including SARIMA, NNAR, TBATS, ETS, and XGBoost, was evaluated using both the training dataset (2018–2023) and the test dataset (2024). Four error metrics were used for model comparison: MAE, MAPE, MASE, and RMSE. The results are presented in Table 1.

On the training dataset, XGBoost produced the lowest errors across all evaluation metrics. It achieved an MAE of 0.02, MAPE of 4.49%, MASE of 0.17, and RMSE of 0.03. While NNAR also performed well on the training set, its performance deteriorated significantly on the test dataset, with a high MAPE of 54.36% and RMSE of 0.21, suggesting a tendency toward overfitting. In contrast, SARIMA, TBATS, and ETS demonstrated relatively consistent performance across both training and test datasets. Among these, TBATS and ETS produced the lowest MASEs and RMSEs in the test dataset, indicating good generalization to unseen data.

Figure 4A provides a visual comparison of the actual %DOA values and the forecasts generated by each model across the full time series from January 2018 to December 2024. Forecasts begin in January 2024, where the model outputs are overlaid on the observed data. Figure 4B shows a zoomed-in view of the period from January 2023 to December 2024, allowing for a more detailed comparison between predicted and actual values. In the test data, the SARIMA, TBATS, ETS, and XGBoost models exhibited approximately similar forecast trends and patterns. In contrast, the NNAR model consistently projected higher %DOA values throughout the entire forecast period.

### 3.3. Forecasting %DOA for 2025

Monthly %DOA forecasts for the year 2025 were generated using the SARIMA, NNAR, TBATS, ETS, and XGBoost models. Figure 5 provides a visual comparison of these forecasts alongside %DOA trends, while the predicted values for each month are detailed in Appendix A.

Figure 5A displays the complete observed %DOA time series from January 2018 through December 2024, with model forecasts for 2025 extending beyond the historical data. Figure 5B presents a zoomed-in view from 2024 to 2025, enabling a more detailed comparison of forecast patterns across models. Most models predicted elevated %DOA during the early months of 2025, followed by a general decline toward mid-year. The TBATS, ETS, and XGBoost models exhibited similar forecast trends. In contrast, the SARIMA and NNAR models projected greater variability, with NNAR showing a consistently higher forecast throughout most of the year.

## 4. Discussion

This study employed time series modeling to investigate trends and seasonal patterns in the %DOA within the ABF broiler production system in Thailand. Forecasts were generated using five models: SARIMA, NNAR, TBATS, ETS, and XGBoost. Each model’s performance was evaluated based on both training and test datasets, enabling the identification of models that offered the most accurate and practical forecasting capabilities.

### 4.1. Interpreting DOA Patterns in ABF Broiler Production

The results revealed a seasonal pattern in %DOA, with consistent peaks in the early months of the year, particularly February, and troughs observed between June and September. Environmental and management factors may influence these seasonal fluctuations during Thailand’s cooler season. During this period, suboptimal ventilation and variable transport conditions may increase physiological stress and contribute to elevated mortality during transit [43,44].

Proper broiler house ventilation is particularly important during cold weather, as it helps maintain thermal comfort and reduces the buildup of harmful gases such as ammonia. Management practices such as reducing fan use at night in evaporative cooling barns, though intended to preserve warmth, can impair air circulation and lead to respiratory issues in broilers [1,45]. Poor respiratory health prior to transport may increase birds’ vulnerability to handling stress and mortality during loading and transit.

Seasonal effects on transport-related mortality have also been reported in other poultry species. For instance, studies in end-of-lay hens have demonstrated increased DOA during colder months due to thermal stress and prolonged transportation conditions [46,47]. While these studies focus on different poultry categories, they provide relevant parallels supporting the observed seasonal effects in broilers.

The Friedman rank sum tests indicated the presence of significant seasonality in the %DOA time series. In addition, the trend component showed a decrease in %DOA between 2018 and 2019, followed by a gradual increase that peaked in 2022, and then declined modestly through 2024. These fluctuations may reflect early operational challenges associated with ABF program implementation and external disruptions such as the COVID-19 pandemic, which affected labor availability, transport logistics, and on-farm monitoring between 2020 and 2022.

Following the pandemic period, the producer implemented improved management strategies that were likely associated with the observed reduction in DOA. Farm-level interventions such as enhanced ventilation systems [48], strict control of loading density, and temperature monitoring during transport [49] contributed to improved broiler survival. Additional biosecurity measures and proactive health monitoring also supported the reduction in mortality rates, despite the absence of antibiotic use.

### 4.2. Model Performance

Among the forecasting models evaluated, TBATS and ETS demonstrated the highest accuracy on the test dataset. These models captured seasonal patterns effectively and exhibited lower error rates, suggesting robust generalizability. Their ability to model both standard and irregular seasonal cycles made them particularly suitable for %DOA forecasting in broiler production.

In contrast, models like NNAR and XGBoost, although showing superior performance on the training dataset, exhibited substantial drops in predictive accuracy when applied to the test dataset. This reduction was particularly evident in the MAPE values, with NNAR reaching over 50% on the test set. High MAPE values indicate that forecast errors, when expressed relative to actual values, were large and variable across months. This could result from the sensitivity of MAPE to small actual values, which may inflate error percentages disproportionately even when absolute differences are modest. In practical terms, such high relative error limits the operational utility of the model for decision-making during critical time periods.

These findings underscore the importance of model stability and interpretability in real-world production settings [14]. While flexible models like NNAR and XGBoost can capture nonlinear relationships in the data, their tendency to overfit the training dataset may hinder their ability to generalize to new data. SARIMA, though less flexible, provided moderate but consistent accuracy, making it a dependable option where model transparency is prioritized. Overall, achieving a balance between complexity, interpretability, and predictive stability is essential when selecting models for routine use in broiler production forecasting.

### 4.3. Forecasting Applications and Future Directions

Forecasted %DOA values for 2025 reflected seasonal trends, with most models predicting peaks in February and declines in the mid- to late-year months. These forecasts have practical relevance, particularly as early warning tools to inform decision-making in broiler production. Anticipating seasonal increases in mortality allows producers to implement targeted interventions, such as adjusting transport timing, modifying crate density, or enhancing pre-shipment management practices during high-risk periods [46,47].

In ABF systems, where disease prevention relies on non-antibiotic strategies, such predictive insights are especially critical. Models such as TBATS and ETS, which demonstrated high forecasting accuracy and seasonal alignment, can be effectively integrated into routine operations. Their application supports proactive planning for transport, handling, and environmental control, thereby enhancing animal welfare, minimizing economic losses, and improving overall production efficiency.

Aligning with the principles of precision livestock farming, time series forecasting enables data-driven decision-making, early intervention, and continuous system improvement [50]. Looking forward, future research should incorporate external explanatory variables such as temperature, humidity, loading density, transport duration, and ventilation conditions to enhance model accuracy [51,52]. Machine learning models like XGBoost, as well as hybrid approaches, may benefit from multivariate inputs that better represent real-time conditions [53]. Broader application of these models across different production systems or geographic regions could improve generalizability and establish industry benchmarks. In the long term, integrating forecasting with sensor-based technologies and automated decision-support tools may offer real-time monitoring and adaptive management in ABF and other antibiotic stewardship programs.

### 4.4. Limitations

Despite its strengths, this study has several limitations. First, it relies on data from a single large broiler producer, which may limit the generalizability of the findings to other operations or geographic regions. Second, the forecasting models used in this study did not incorporate external covariates such as temperature, humidity, loading time, loading density, transport duration, or flock characteristics, all of which could influence %DOA and potentially enhance prediction accuracy. Third, the performance of time series models may be affected by unanticipated events such as disease outbreaks or regulatory changes, which are difficult to capture using past data alone.

## 5. Conclusions

This study identified seasonal trends and forecasted %DOA in broilers raised under the ABF program using time series models. Among the evaluated models, TBATS and ETS demonstrated the highest forecasting accuracy. Additionally, SARIMA, TBATS, ETS, and XGBoost produced aligned forecast trends for %DOA in 2025. These findings suggest that time series forecasting can serve as a valuable tool for broiler producers, supporting proactive management strategies to reduce mortality risks, improve animal welfare, and enhance operational efficiency in ABF production systems.

## Figures and Tables

**Figure 1 animals-15-01179-f001:**
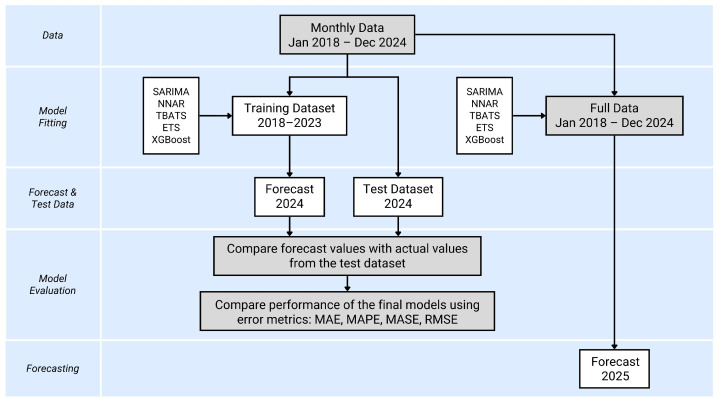
The time series modeling procedure, illustrating the dataset splitting into training and test datasets. The models used in this study include Seasonal AutoRegressive Integrated Moving Average (SARIMA), Neural Network AutoRegressive (NNAR), Trigonometric Box-Cox ARMA Trend Seasonal (TBATS), Exponential Smoothing State Space (ETS), and Extreme Gradient Boosting (XGBoost). Model performance was evaluated using four error metrics: mean absolute error (MAE), mean absolute percentage error (MAPE), mean absolute scaled error (MASE), and root mean square error (RMSE).

**Figure 2 animals-15-01179-f002:**
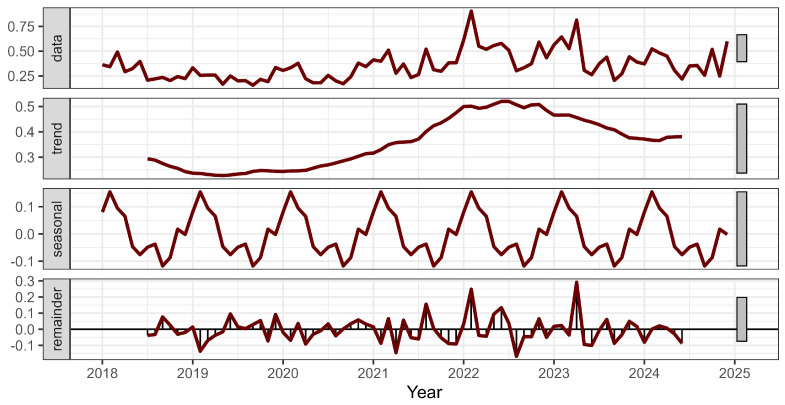
Time series decomposition of monthly %DOA in broilers raised under the ABF program from 2018 to 2024. The plot illustrates the original data (top), a gradual upward trend until 2023 followed by a slight decline in 2024, and consistent seasonal patterns that repeat annually. The remainder component (bottom panel) represents random or irregular variations not explained by the trend or seasonal structure.

**Figure 3 animals-15-01179-f003:**
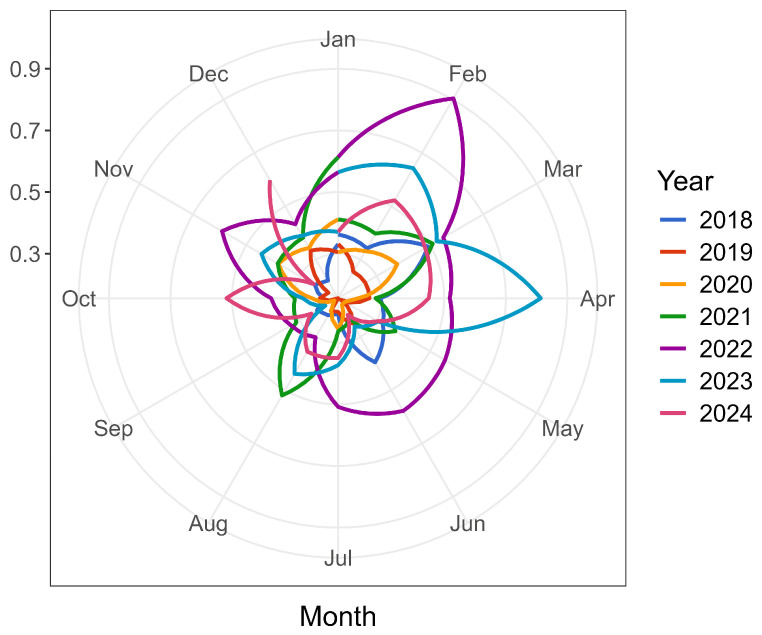
Circular seasonal plot of monthly %DOA from 2018 to 2024, showing recurring annual patterns with variation in peak months across years.

**Figure 4 animals-15-01179-f004:**
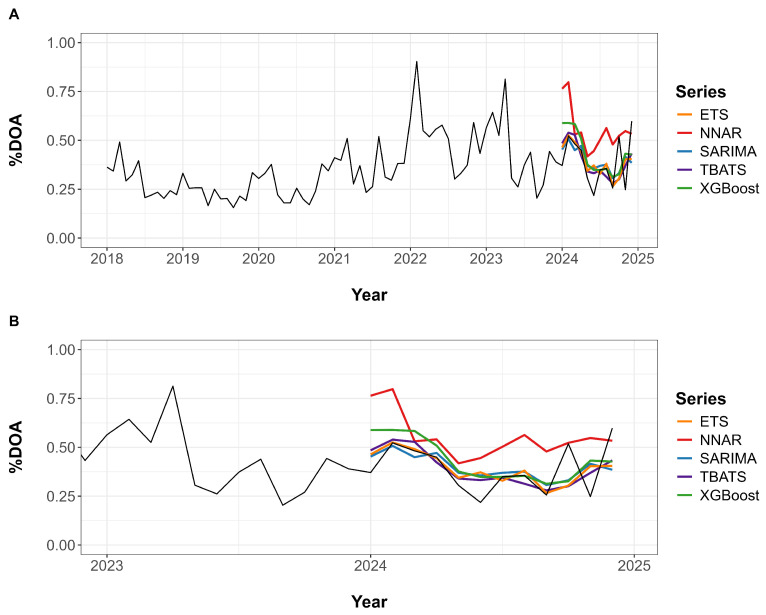
Forecasting performance of five time series models for broiler %DOA. (**A**) Full time series from 2018 to 2024, displaying actual %DOA values alongside model forecasts starting in 2024. (**B**) A zoomed-in view from 2023 to the end of 2024, highlighting differences between observed values and model predictions.

**Figure 5 animals-15-01179-f005:**
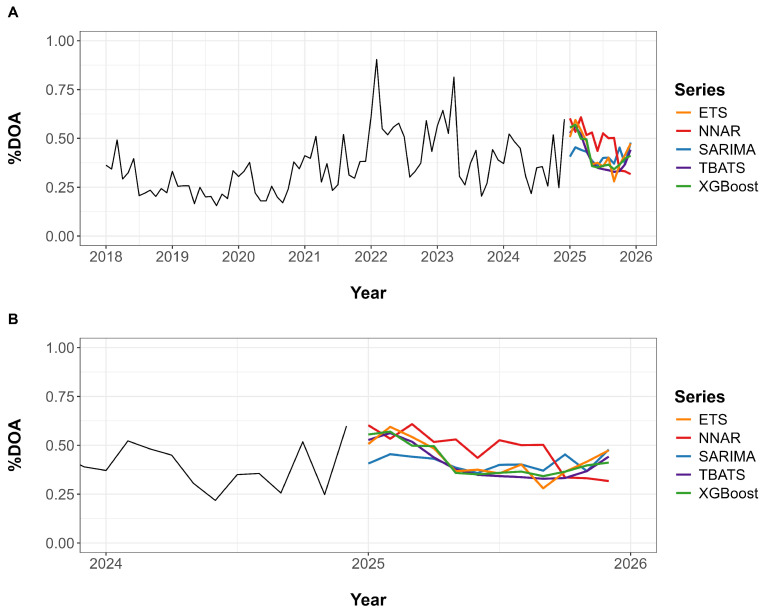
Forecasts of monthly %DOA in 2025 using the SARIMA, NNAR, TBATS, ETS, and XGBoost models. (**A**) Full time series from January 2018 to December 2024, including forecasted values for 2025. (**B**) A zoomed-in view from 2024 to the end of 2025 for clearer comparison among model forecasts.

**Table 1 animals-15-01179-t001:** Forecast error metrics for broilers’ %DOA using training (2018–2023) and test (2024) datasets across five time series models.

Model	Training Dataset				Test Dataset			
	MAE	MAPE	MASE	RMSE	MAE	MAPE	MASE	RMSE
SARIMA	0.08	23.74	0.81	0.11	0.08	24.30	0.59	0.11
NNAR	0.04	11.24	0.44	0.06	0.18	54.36	1.22	0.21
TBATS	0.07	18.35	0.68	0.10	0.08	21.22	0.54	0.10
ETS	0.07	20.04	0.70	0.10	0.08	22.11	0.54	0.11
XGBoost	0.02	4.49	0.17	0.03	0.10	29.33	0.73	0.13

SARIMA: AutoRegressive Integrated Moving Average; NNAR: Neural Network AutoRegressive; TBATS: Trigonometric Box-Cox ARMA Trend Seasonal; ETS: Exponential Smoothing State Space; XGBoost: Extreme Gradient Boosting; MAE: mean absolute error; MAPE: mean absolute percentage error; MASE: mean absolute scaled error; RMSE: root mean square error.

## Data Availability

The data supporting the findings of this study are available from the corresponding author upon reasonable request. Restrictions apply to the availability of these data due to confidentiality agreements with the data provider.

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
