# Peer review of "Modeling and Forecasting Dead-on-Arrival in Broilers Using Time Series Methods: A Case Study from Thailand"

_animals, 2025, doi:10.3390/ani15081179_

Round 1
Reviewer 1 Report
Comments and Suggestions for Authors
This article was written appropriately and justice was made to the title. However, the introduction speaks more to Thailand broiler production. Whereas the title did not talk about Thailand. I would suggest that the author insert Thailand in the title to be more specific.
Author Response
Point 1: This article was written appropriately and justice was made to the title. However, the introduction speaks more to Thailand broiler production. Whereas the title did not talk about Thailand. I would suggest that the author insert Thailand in the title to be more specific.
Response 1: Thank you for your helpful suggestion. We agree that including the geographic context in the title improves clarity and aligns better with the content of the introduction and dataset. Accordingly, we have revised the title to:
“Modeling and Forecasting Dead-on-Arrival in Broilers Using Time Series Methods: A Case Study from Thailand.”
Reviewer 2 Report
Comments and Suggestions for Authors
The manuscript is well written and has a clear rationale for the study. The authors identified the real-world issue (% DOA) and logically led the time series forecasting. However, I would like to suggest some modifications for enhancing the manuscript, which includes:
- Please consider revising the title for this manuscript, as the study is focused on time series methodology rather than the “Antibiotic-Free Production System” itself. The seasonal patterns identified suggest the environmental factors may be more influential rather than the infection status (without direct evidence, this may be speculative).
- In Figure 2 (bottom panel), please consider writing about the remainder component so that readers can understand it.
- Suggestion to discuss more about high MAPE values, why they are high, and what this indicates about the forecast quality in practical operational terms.
Author Response
Point 1: Please consider revising the title for this manuscript, as the study is focused on time series methodology rather than the “Antibiotic-Free Production System” itself. The seasonal patterns identified suggest the environmental factors may be more influential rather than the infection status (without direct evidence, this may be speculative).
Response 1: Thank you for your insightful feedback. We acknowledge that the primary focus of this study is the application of time series models for forecasting %DOA, rather than an evaluation of the antibiotic-free (ABF) production system itself. To reflect this methodological focus and reduce potential misinterpretation, we have revised the title to:
“Modeling and Forecasting Dead-on-Arrival in Broilers Using Time Series Methods: A Case Study from Thailand.”
This revision more accurately describes the study’s scope and avoids overstating the role of ABF as the central research question.
Point 2: In Figure 2 (bottom panel), please consider writing about the remainder component so that readers can understand it.
Response 2: Thank you for your helpful suggestion. We have revised the caption for Figure 2 to include a clear explanation of the remainder (residual) component. This component represents irregular variations not captured by the trend or seasonality. The updated caption now clarifies this interpretation for readers. (Line 295-299)
Point 3: Suggestion to discuss more about high MAPE values, why they are high, and what this indicates about the forecast quality in practical operational terms.
Response 3: We appreciate the reviewer’s insightful comment regarding the high MAPE values. We have revised the “Model Performance” section of the Discussion (Section 4.2) to provide a clearer interpretation of MAPE in the context of model evaluation. The revised text discusses how high MAPE values, especially for NNAR and XGBoost, may result from overfitting and the metric’s sensitivity to small actual values. We also expanded the discussion on the practical implications of high MAPE values and how they affect model reliability in operational settings. These revisions help clarify the trade-offs between forecast accuracy, model complexity, and practical applicability. Thank you for helping us improve the clarity and utility of the manuscript. (Line 401-409)
Reviewer 3 Report
Comments and Suggestions for Authors
Dear authors,
Some points request your attention, such as :
C1 Regarding the reference 42, line 370 - it is in relation with stress transport of end-of-lay hens
C2- Regarding the reference 43, line 371 - the study of Di Martino et al. includes also the endogenous factors associated with pre-slaughter mortality in turkeys and end of lay hens
C2 Regarding the reference 45 - according to study of Van Limbergen et al. (2020), the transport details after loading the broilers were not collected.
C3 - Regarding the reference 49, line 430 - the study is focused on mortality difference between hybrids, but not include data of transport duration.
Author Response
Point 1: Regarding the reference 42, line 370 - it is in relation with stress transport of end-of-lay hens
Response 1: Thank you for pointing this out. We acknowledge that Reference 42 focuses on transport-related stress in end-of-lay hens. While this study does not focus on broilers, we cited it to illustrate the broader impact of environmental and seasonal stressors on poultry during transportation. In the revised text, we have clarified that the reference serves as comparative evidence from a related poultry context rather than direct findings in broilers. In the updated manuscript, it is Reference 46. (Line 378-380)
Point 2: Regarding the reference 43, line 371 - the study of Di Martino et al. includes also the endogenous factors associated with pre-slaughter mortality in turkeys and end of lay hens.
Response 2: We appreciate the comment regarding Reference 43. The study by Di Martino et al. includes both endogenous and exogenous factors related to pre-slaughter mortality in turkeys and end-of-lay hens. We have updated the manuscript to clarify that this reference is used to support the relevance of seasonal and management-related influences on preslaughter mortality, while acknowledging species-specific and physiological differences. This adjustment ensures that the discussion does not overstate generalizability and remains appropriately cautious in interpretation. In the updated manuscript, it is Reference 47. (Line 378-380)
Point 3: Regarding the reference 45 - according to study of Van Limbergen et al. (2020), the transport details after loading the broilers were not collected.
Response 3: We appreciate the reviewer’s observation. We have reviewed the study by Van Limbergen et al. (2020) and agree that transport conditions after loading were not covered in the data collection. The reference was initially cited after the mention of “temperature monitoring during transport,” which may have caused confusion. To address this, we have revised the sentence and relocated the citation so that it now refers solely to “enhanced ventilation systems,” which were directly addressed in the referenced study. This correction ensures a more accurate attribution of the supporting evidence and maintains the integrity of the discussion. (Reference 48, Line 391)
Point 4: Regarding the reference 49, line 430 - the study is focused on mortality difference between hybrids, but not include data of transport duration.
Response 4: We thank the reviewer for the helpful observation. While the study by Forseth et al. (2024) primarily compared DOA rates between fast- and slow-growing broiler strains, it did include findings related to environmental conditions during transport, particularly highlighting increased DOA under colder temperatures. These insights are tangentially related to the discussion of seasonal influences and ambient conditions affecting transport stress.
However, we agree that the reference does not directly address transport duration or most of the specific factors listed in the sentence. To avoid overclaiming and ensure that each reference accurately supports the content, we have decided to remove this citation from the revised manuscript. (Line 435, the reference was removed.)
Reviewer 4 Report
Comments and Suggestions for Authors
Time Series Analysis and Forecasting of Dead-on-Arrival in Broilers Raised under an Antibiotic-Free Production System
Specific Comments
Line 37-38: " Among the evaluated models, TBATS and ETS demonstrated the highest forecasting accuracy when applied to the test data.”
Comment: “How much higher was the forecasting accuracy?”.
Suggestions: “Mention clearly the difference in forecasting accuracy you found while comparing models.”
Line 20-22: " Detailed protocols for broiler rearing, handling, transportation, and %DOA calculation have been previously described.”
Comment: “Can you write these protocols as a separate section of the methods?”
Suggestions: “This will give good flow while reading the methods section.”
For Figure 3 and 4, use the color seems too bright. Can you use lightened color palettes?
Line 49, 52, 112, 142: Provide references for relevant information
General Comments
- Clarity: The writing and flow of the manuscript seems clear to me.
- Methodology: Methodology seems well written.
- Results: Results were presented well by authors.
Comments on the Quality of English Language
Can be improved
Author Response
Point 1: Line 37-38: “Among the evaluated models, TBATS and ETS demonstrated the highest forecasting accuracy when applied to the test data.”
Comment: “How much higher was the forecasting accuracy?”.
Suggestions: “Mention clearly the difference in forecasting accuracy you found while comparing models.”
Response 1: Thank you for this valuable suggestion. We have revised the Abstract to include specific MAPE values for the top-performing models (TBATS and ETS) and provided a comparison with NNAR and XGBoost to highlight the relative differences in forecasting accuracy. While multiple metrics were used to evaluate model performance, we selected MAPE for presentation in the Abstract because it offers an interpretable percentage-based measure of accuracy. The full manuscript retains comprehensive performance comparisons using all four metrics (MAE, MAPE, MASE, and RMSE) in both the Results and Discussion sections. (Line 36-39)
Point 2: Line 20-22: “Detailed protocols for broiler rearing, handling, transportation, and %DOA calculation have been previously described.”
Comment: “Can you write these protocols as a separate section of the methods?”
Suggestions: “This will give good flow while reading the methods section.”
Response 2: To improve the clarity and logical flow of the Methods section, we have revised the manuscript by creating a new subsection titled “2.2. Rearing, Handling, and Transportation Protocols.” This subsection now presents the relevant management protocols separately from the data collection description. The Data Collection details remain in Section 2.1, and no new information has been added beyond what was previously described. We believe this structural revision improves readability and aligns with your recommendation. (Line 102-128)
Point 3: For Figure 3 and 4, use the color seems too bright. Can you use lightened color palettes?
Response 3: Thank you for your helpful suggestion. Figures 3 and 4 have been revised by adjusting the color palettes to use darker and more visually accessible tones, which improves contrast and readability. Additionally, Figure 5, which presents the model predictions for the year 2025, has also been updated to ensure consistent color usage with Figure 4. All revised figures have been included in the updated manuscript.
Point 4: Line 49, 52, 112, 142: Provide references for relevant information
Response 4: We have provided appropriate references to support the information at the indicated locations. The updated references have been added to the revised manuscript at Lines 50, 53, 112, and 140, as suggested.